# Artificial Intelligence and Radiologists at Prostate Cancer Detection in MRI – The PI-CAI Challenge

**Anindo Saha**[*1]                    ANINDYA.SHAHA@RADBOUDUMC.NL
**Joeran S. Bosma**[*1]                JOERAN.BOSMA@RADBOUDUMC.NL
**Jasper J. Twilt**[*1]                JASPER.TWILT@RADBOUDUMC.NL
**Bram van Ginneken**[1,2]             BRAM.VANGINNEKEN@RADBOUDUMC.NL
**Derya Yakar**[3,4]                   D.YAKAR@UMCG.NL
**Mattijs Elschot**[5,6]               MATTIJS.ELSCHOT@NTNU.NO
**Jeroen Veltman**[7]                  J.VELTMAN@ZGT.NL
**Jurgen Fütterer**[1]                 JURGEN.FUTTERER@RADBOUDUMC.NL
**Maarten de Rooij**[†1]               MAARTEN.DEROOIJ@RADBOUDUMC.NL
**Henkjan Huisman**[†1,5]              HENKJAN.HUISMAN@RADBOUDUMC.NL

[1] *Department of Medical Imaging, Radboud University Medical Center, The Netherlands*

[2] *Fraunhofer Institute for Digital Medicine MEVIS, Germany*

[3] *Department of Radiology, Nuclear Medicine and Molecular Imaging, University Medical Center Groningen, The Netherlands*

[4] *Department of Radiology, Netherlands Cancer Institute, The Netherlands*

[5] *Department of Circulation and Medical Imaging, Norwegian University of Science and Technology, Norway*

[6] *Department of Radiology and Nuclear Medicine, St. Olavs Hospital, Trondheim University Hospital, Norway*

[7] *Department of Radiology, Ziekenhuis Groep Twente, The Netherlands*

**Editors:** Under Review for MIDL 2023

## Abstract

We hypothesized that state-of-the-art AI models, trained using thousands of patient cases, are non-inferior to radiologists at clinically significant prostate cancer diagnosis using MRI. To test the same, we designed an international comparative study titled the PI-CAI challenge, where we investigated AI models that were independently developed, trained and externally tested using a large multi-center cohort of 10,207 patient exams. Preliminary results indicate that when trained on 1,500 cases only, such models already achieve diagnostic performance comparable to that of radiologists reported in literature.

**Keywords:** prostate cancer, artificial intelligence, magnetic resonance imaging, radiologists, computer-aided detection and diagnosis

## 1. Introduction

Clinically significant prostate cancer (csPCa) caused over 375,000 deaths worldwide in 2020 (Sung et al., 2021). Magnetic resonance imaging (MRI) is playing an increasingly important

---

[*] Contributed equally

[†] Contributed equally

role in csPCa management, and has been recommended by recent clinical guidelines in the European Union, United Kingdom and the United States (Mottet et al., 2021; NICE, 2019; Eastham et al., 2022). Artificial intelligence (AI) algorithms have matched expert clinicians in medical image analysis across several domains, and can address the rising demand in imaging (Milea et al., 2020; Bulten et al., 2022; McKinney et al., 2020; Hricak et al., 2021). However, limited scientific evidence on the efficacy of AI-driven csPCa diagnosis impedes its widescale adoption (van Leeuwen et al., 2021; Angus, 2020). We hypothesized that state-of-the-art AI models, trained using thousands of patient cases, are non-inferior to radiologists at csPCa diagnosis using MRI. To test the same, we designed an international comparative study, titled the PI-CAI challenge (https://pi-cai.grand-challenge.org/).

## 2. Materials and Methods

The PI-CAI study protocol was established in conjunction with 16 experts across prostate radiology, urology and AI (Saha et al., 2022). This retrospective study included 10,207 prostate MRI exams (9,129 patients) curated from four European tertiary care centers between 2012–2021. All patients were men suspected of harboring prostate cancer, without a history of treatment or prior csPCa findings. Imaging was acquired using various commercial 1.5 or 3T MRI scanners, equipped with surface coils. In the first phase of this study, algorithm developers worldwide were invited to design AI models for detecting csPCa in biparametric MRI (bpMRI), using 1,500 training cases that were made publicly available. For a given bpMRI exam, AI models were required to complete two tasks: localize all csPCa lesions (if any), and predict the case-level likelihood of csPCa diagnosis. To this end, AI models could use imaging data and several variables (PSA, patient age, prostate volume, scanner model) to inform their predictions. Once developed, these algorithms were independently tested using a hidden cohort of 1,000 patient cases (including external data from an unseen center) in a fully-blinded setting, where histopathology and a follow-up period of $\geq 3$ years were used to establish the reference standard.

## 3. Results and Conclusion

Between June–November 2022, >830 AI developers (>50 countries) opted-in and >310 algorithm submissions were made. Parallel to this, 79 radiologists (55 centers, 22 countries) enlisted in a multi-reader multi-case observer study, whose primary objective was to estimate clinician's performance at this same task. Distribution of AI developers and radiologists has been illustrated in Figure 1. When trained on 1,500 cases, the top five most performant prostate-AI models reached 0.88±0.01 AUROC in case-level diagnosis, and 76.38±0.74% sensitivity at 0.5 false positives per case in lesion detection (as shown in Table 1), which is comparable to that of radiologists' performance reported in literature (Schelb et al., 2019; Hosseinzadeh et al., 2022; Roest et al., 2023). When ensembled with equal weighting, diagnostic performance increased substantially to 0.912 AUROC, indicating notable diversity among the top five methods. In the next phase of the challenge, these AI models will be re-trained using a private dataset of 9,107 cases, performance will be re-evaluated across 1,000 testing cases, and the ensembled AI system will be benchmarked against radiologists participating in the reader study and the historical reads made during routine practice.

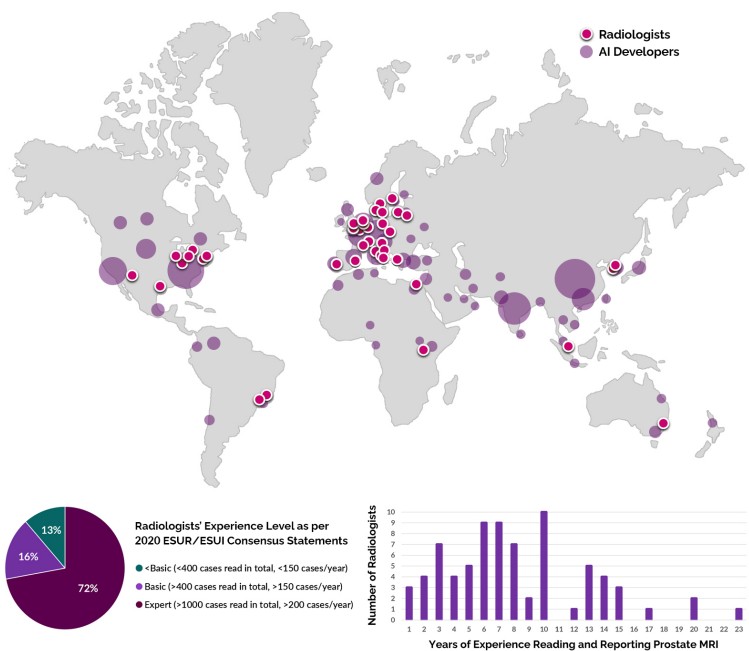

Figure 1: Distribution of > 830 AI developers (> 50 countries) and 79 radiologists (55 centers, 22 countries) participating in the PI-CAI challenge, as of 10 November, 2022. Radiologists' experience varies between 1 and 23 years (median 7 years), where 72% (57) of readers can be categorized as "expert" based on the 2020 ESUR/ESUI consensus statements (de Rooij et al., 2020).

Table 1: Case-level diagnostic performance, as estimated by the *Area Under Receiver Operating Characteristic* (AUROC) metric, and lesion-level detection performance, as estimated by the *Average Precision* (AP) and the detection sensitivity at 0.5 false positives per patient metrics, across 1,000 testing cases.

| Model | AUROC | AP | Sens @ 0.5 FP/Patient |
|---|---|---|---|
| **Y. Yuan et al.** (Australia) | 0.881 | 0.633 | 77.64% |
| **C. A. Nader et al.** (France) | 0.889 | 0.615 | 76.63% |
| **A. Karagöz et al.** (Turkey) | 0.889 | 0.614 | 75.38% |
| **X. Li, S. Vesal, S. Saunders** et al. (USA) | 0.871 | 0.612 | 76.13% |
| **H. Kan et al.** (China) | 0.886 | 0.593 | 76.13% |
| **Ensemble of Top Five Models** (Global) | 0.912 | – | – |

## Acknowledgments

This study has been endorsed by MIDL, MICCAI, the European Society of Urogenital Radiology, the European Association of Urology, and supported in parts by Amazon Web Services, EU H2020: ProCAncer-I and Health~Holland.

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
