# OpenReview forum: "Artificial Intelligence and Radiologists at Prostate Cancer Detection in MRI — The PI-CAI Challenge"
_MIDL.io/2023/Short_Paper_Track — MIDL 2023 Short paper track Poster_

### Official Review · Reviewer_hFey · 2023-04-17
**This work presents the PI-CAI challenge, a research initiative that aims to evaluate the performance of independently developed AI models on a large and diverse dataset of patient exams for prostate cancer detection. The main goal of the challenge is to identify AI models that are effective in accurately detecting and diagnosing medical conditions, and have potential for use in clinical practice.**

**Rating:** 8
**Confidence:** 4

**Review:**

The paper presents a summary of the results of the PI-CAI challenge, which evaluated the performance of AI models developed by five independent research teams, as well as an ensemble of all five methods. The paper is clearly written.

**Pros:**
- Large and diverse dataset: This work used a multi-center cohort of 10,207 patient exams, which is a substantial and diverse dataset that can help to ensure the generalizability of the results.

- Independent model development: The AI models used in the challenge were independently developed by different research teams, which can help to reduce the risk of bias and increase the reliability of the results.

- External testing: The models were externally tested on the dataset, which can provide a more accurate evaluation of their performance than internal testing.

- Clinical relevance: The challenge focused on the detection and diagnosis of prostate cancer on npMRI, which is a clinically relevant task that has important implications for patient outcomes. The results could be interesting for the MIDL community.

**Cons:**
- The authors did not present the AP and Sensitivity scores, in addition to the AUC score, for the ensemble of all 5-teams, which could limit the comprehensiveness of the evaluation of the model's performance.

- It is likely that most of the participants had an ensemble model for their submission, and the authors reported an AUC of 91.2% for the ensemble of all models. However, it is unclear how long it would take for the final model to predict prostate cancer and its likelihood. This raises questions about the clinical feasibility of using such a large model in terms of resources and time.

- It is not clear from the information provided whether the authors have commented on the next steps of the PI-CAI challenge, including whether the final trained model or code will be made available for the community to test, train, and fine-tune these models.

---

### Official Review · Reviewer_9KEx · 2023-04-20
**AI matches radiologists in prostate cancer diagnosis, more detail on lesion-level evaluation and top 5 algorithms needed**

**Rating:** 7
**Confidence:** 4

**Review:**

The paper suggests that AI models trained on patient cases can match radiologists' ability in diagnosing prostate cancer using MRI. In the PI-CAI challenge study, which included 10,207 patient exams, AI models trained on 1,500 cases demonstrated diagnostic performance comparable to radiologists. This research is significant for the field of AI and MRI-based prostate cancer diagnosis.

It would have been helpful if the authors had provided more details on how to evaluate the performance at the lesion-level. Additionally, a brief description of each of the top 5 performance algorithms would have added value to the paper.